# Mechanisms of Formation of Antibodies against Blood Group Antigens That Do Not Exist in the Body

**DOI:** 10.3390/ijms242015044

**Published:** 2023-10-10

**Authors:** Alexander A. Mironov, Maksim A. Savin, Anna V. Zaitseva, Ivan D. Dimov, Irina S. Sesorova

**Affiliations:** 1Department of Cell Biology, IFOM ETS—The AIRC Institute of Molecular Oncology, Via Adamello, 16, 20139 Milan, Italy; 2The Department for Welding Production and Technology of Constructional Materials, Perm National Research Polytechnic University, Komsomolsky Prospekt, 29, 614990 Perm, Russia; savin.ma@yandex.ru; 3Department of Anatomy, Saint Petersburg State Pediatric Medical University, 194100 Saint Petersburg, Russia; 4Department of Anatomy, Ivanovo State Medical Academy, 153012 Ivanovo, Russia

**Keywords:** Golgi complex, Golgi apparatus, endothelial cells, enterocytes, glycosylation, intracellular transport, blood group antibody, glycosylation mistakes, tolerance

## Abstract

The system of the four different human blood groups is based on the oligosaccharide antigens A or B, which are located on the surface of blood cells and other cells including endothelial cells, attached to the membrane proteins or lipids. After transfusion, the presence of these antigens on the apical surface of endothelial cells could induce an immunological reaction against the host. The final oligosaccharide sequence of AgA consists of Gal-GlcNAc-Gal (GalNAc)-Fuc. AgB contains Gal-GlcNAc-Gal (Gal)-Fuc. These antigens are synthesised in the Golgi complex (GC) using unique Golgi glycosylation enzymes (GGEs). People with AgA also synthesise antibodies against AgB (group A [II]). People with AgB synthesise antibodies against AgA (group B [III]). People expressing AgA together with AgB (group AB [IV]) do not have these antibodies, while people who do not express these antigens (group O [0; I]) synthesise antibodies against both antigens. Consequently, the antibodies are synthesised against antigens that apparently do not exist in the body. Here, we compared the prediction power of the main hypotheses explaining the formation of these antibodies, namely, the concept of natural antibodies, the gut bacteria-derived antibody hypothesis, and the antibodies formed as a result of glycosylation mistakes or de-sialylation of polysaccharide chains. We assume that when the GC is overloaded with lipids, other less specialised GGEs could make mistakes and synthesise the antigens of these blood groups. Alternatively, under these conditions, the chylomicrons formed in the enterocytes may, under this overload, linger in the post-Golgi compartment, which is temporarily connected to the endosomes. These compartments contain neuraminidases that can cleave off sialic acid, unmasking these blood antigens located below the acid and inducing the production of antibodies.

## 1. Introduction

The main ABO blood groups were discovered by Landsteiner [1]. The molecular and genetic mechanisms of the blood group antigens had been discovered by Yamamoto et al. [2]. Briefly, according to the “Landsteiner’s Law”, people with the group O (although known in some countries as 0, it is internationally known as O) do not synthesise the antigens A and B (AgA, AgB), but have antibodies (Ab) against AgA (AbA) and AgB (AbB). Individuals with the blood group A produce AgA and AbB. People with the blood type B synthesise AgB and AbA. Finally, people with the blood type AB synthesise AgA and AgB but they do not have AbA and AbB. People of the so-called Bombay group (H-null Ag) cannot synthesize AgA and AgB. Now the situation becomes even more complicated [3,4].

The PM of erythrocytes contains about one million molecules of glycosphingolipids with AgA and AgB. In addition to blood cells, AgA and AgB are expressed in neurons, vascular endothelium, and on the basolateral plasma membrane (BLPM) of some epithelial cells, for example, nephrocytes, enterocytes, etc. [5,6]. Knowledge about the mechanisms of formation of these blood groups is extremely important not only for haematology (in cases of haemolytic disease of the foetus and newborn), blood transfusion, but also for transplantation since the above antibodies can cause the hyperacute rejection of the graft. Removal of the blood group antibodies against a donor organ before transplantation incompatible with ABO can prevent episodes of hyper-acute rejection [7,8].

The main paradox of this field is the fact that people who do not have an antigen form antibodies against it; there is a contradiction between the absence of the antigens in the body and the impossibility to absorb these antigens from the intestine. The mechanism responsible for the formation of the antibodies against the human blood group polysaccharide (PS) antigens is unknown. Branch [9] claimed that “the origin of the ABO isohaemagglutinins was not really resolved”. In the most recent article by Jajosky et al. [4] state that there is uncertainty about the origin of the so-called ‘natural antibodies’. According to Jajosky et al. [4], there are conflicting data regarding the potential role of plant raw materials, microbes, and genetic elements.

The presence of AbA and AbB are important for transplantation. ABO compatibility has, therefore, been required for successful cadaveric transplantation. The endothelial cells (ECs) of the allograft contain ABO Ags. The donor-specific antibodies could cause allograft rejection [10,11,12,13,14,15].

Complement-fixing preformed donor-specific Abs are associated with a poorer prognosis compared to non-fixing Abs [16]. These Ags are carbohydrates linked to glycoproteins and glycolipids. Importantly, AbA and AbB are formed in people lacking AgA and AgB [13]. ECs express a lot of AgA and AgB, especially on their luminal surface [16,17]. Upon binding to A/B-incompatible ECs, AbA and AbB cause hyper-acute or accelerated acute graft rejection [18]. The Ags associated with the blood groups A and H are found in normal and neoplastic epithelium of the bladder [19]. AbA and AbB are expressed on the APM of ECs and on the erythrocyte membrane; they are available for AbA and AbB, and these clones are destroyed.

The origin of AbA and AbB is not really resolved. Understanding of the mechanisms of the formation of AbA and AbB is important for the prevention of organ rejection. AgA and AgB can be effectively removed ex vivo prior to implantation. Also, it is important to remember that blood transfusion is a moderate risk factor for the embolism of lung arteries. During plasma and blood transfusion, there may be a reaction of the donor against the recipient. This is indirectly evidenced by observations that blood transfusion is a risk factor for the embolism of lung arteries [20].

Transplantation of the liver, in which there is a lot of blood inside its sinuses, is also dangerous. The issue has not been well-studied. Thus, this issue deserves our examination. Here, being experts in the Golgi complex (GC) and ECs, we present new hypotheses for their production based on our own data. Since there are thousands of reviews and original papers, we quoted only those which are important for our task and only very recent reviews. We beg our pardons from the scientists which have not been quoted.

## 2. Structure and Origin of the A and B Antigens

The antigens of the blood groups of ABO systems are carbohydrate. Both AgA and AgB are localised on the ends of their oligosaccharides. AgA and AgB appear as GalNAca1,3(Fucα1,2) Gal- and Galα 1,3(Fucα1,2)Gal-, respectively [21]. The blood group AB individuals express both the A and B enzymes able to form AgA and AgB. The individuals with the so-called ‘‘Bombay’’ phenotype lack both the type 1 and type 2 H antigens [22]. AgH is required for synthesizing AgA and AgB, whereas individuals with the Bombay phenotype do not generate the A or B structures (reviewed in [4]). These oligosaccharides have short branches (each being composed of only one monosaccharide) forming “short forks”. The “short forks” in AgA and AgB are composed of fucose and galactose or N-acetyl-galactosamine (GalNAc), being attached to the oligosaccharide chain via GalNAc [4,22,23]. The presence of AgA and AgB on the apical surface of endothelial cells covered with glycocalyces suggests that AgA and AgB could be formed on the end of both oligosaccharides and PSs. Another fork can be attached to AgA, which leads to two fucose residues being attached to two galactoses located through one GalNAc. They are exposed laterally to the chain [4]. Figure 1 shows the molecular structure of AgH, AgA, and AgB, and possible variations.

Usually, AgA and AgB are oligosaccharides. The presence of AgA and AgB on the apical surface of endothelial cells covered in glycocalyces suggests that AgA and AgB could be formed on the end of PSs. These Ags are present in all human cells [14,24]. In the endoplasmic reticulum (ER), the primary polysaccharide chain taken from the dolichol protein is attached to proteins and lipids. AgH is synthesized by epithelial cells and is a precursor of the soluble antigens that are found in the saliva or plasma in “secretory” individuals. The synthesis of AgA and/or AgB leads to a decrease in the amount of H antigen on the surface of the erythrocyte. Thus, the expression of AgA and AgB and the expression of AgH are inversely proportional [25].

In contrast to the process of the formation of antibodies against foreign antigens, which occurs after the presentation of these antigens on macrophages, antibodies to the blood group antigens are formed spontaneously during the first few months of life. Concentration of the blood group antigens in the erythrocytes of newborns can reach 50% of the level of an adult person. With age, AgA and AgB become more and more branched. In the same individual, different tissues express AgB and AgA differently [4,23,26]. Importantly, there are several unclear questions, namely, it is not clear how the maturation of the erythrocyte occurs and whether the PM of erythrocytes is derived from the APM or BLPM. Erythrocytes receive lipids and proteins with AgA and AgB from macrophages. In erythroblasts the GC is very small [27]. It is possible that when the erythrocyte matures, sialylated AgA and AgB are not formed and then they immediately go to the BLPM. Erythrocytes can capture membranes with glycolipids in macrophages.

A and B antigens are also expressed by epithelial cells and ECs. There is AgA and AgB even in semen. If there is a glycocalyx on the EC, then the proteins get there only if it is an APM. After sialylation in an acidic environment, the hydrogen bonds between the sialic acids induce the formation of polysaccharide aggregates, which are transported at the APM [21]. AgA and AgB are recognised by immune cells [27,28,29].

## 3. Glycosylation of Proteins and Lipids in the Golgi Complex by Golgi Glycosylation Enzymes Synthesizing AgA and AgB

Human glycosylation is based on the combination of only several monosaccharides glucose (Glc), galactose (Gal), N-acetylglucosamine (GlcNAc), N-acetyl-galactosamine (GalNAc), glucuronic acid, iduronic acid (IdoA), xylose, mannose, fucose (Fuc), sialic acid N-acetylneuraminic acid (NeuAc) [30,31], and (rarely) N-glycolylneuraminic acid (NeuGc) [32,33,34]. Fuc is the only deoxyhexose [35], and NeuAc and NeuGc are the only sialic acids [36] found on human glycoconjugates. Despite the recognition of mono- and disaccharide epitopes, most circulating carbohydrate-specific Abs bind with a low specificity to larger glycoconjugates [37,38].

The vast majority of glycans on the surface of the PM have sialic acid on their termini. Galactose is always masked with sialic acid or another monosaccharide. The chain can end in glucose, mannose, N-acetyl-galactosamine, N-acetylglucosamine, and glucuronic acid. Usually, fucose attaches to galactose and masks it. There may be a fork on mannose and N-acetyl-galactosamine. Galactose can be attached to glycosphingolipids after glucose and a fork with fucose and GalNAc appears on it. The latter is closed by galactose, which is then covered by fucose [4,26].

Glycosylation of the membrane secretory proteins and some lipids occurs in the GC. In mammalian cells, the GC consists of three–eight cisternae that look like flat discs or flat tubules. These cisternae form stacks. In the transporting stacks, their inner side is covered with the highly perforated cis-most cisterna. Their trans-side is covered with the highly perforated trans-most common cisterna, which is connected to a tubular trans-Golgi network consisting of tubules or flat cisternae and possessing the features of the endosomal system. The edges of the Golgi cisternae, including the CMC, contain coatomer I (COPI)-coated buds. The TMC contains clathrin-coated buds. There are many COPI-dependent 52-nm membrane vesicles in the resting Golgi stacks. On the trans-side of the Golgi stacks, uncoated clathrin-dependent vesicles or these vesicles covered with clathrin can be observed. The Golgi glycosylation enzymes are present in the Golgi cisternal membranes and are localized in accordance with the consequences of the glycosylation process. The transport through the GC is based on the “kiss and run” transport model [39,40,41,42,43,44,45]. Several Golgi enzymes are involved in the production of AgA and AgB. When one Golgi enzyme happens to lose its function, the precursor carbohydrate structure (H) will accumulate without further modification [38].

The resident Golgi glycosylation enzymes (GGEs) transfer sugar residues from an activated donor substrate, usually a nucleotide sugar, to an acceptor, which can be a lipid, protein, or a growing oligosaccharide. The GGEs responsible for the synthesis of AgA and AgB are GalNAc-transferase (GTA) and Gal-transferase (GTB). GTA and GTB differ in their sequence by only four amino acids [2].

There are thousands of known alleles of these genes. In any case, there are three main alleles. Due to the combinatorial nature of glycans there are dozens of glycans that have these antigens on their ends. For example, the O alleles can produce an inactive protein. These genes are present on Chromosomes 9 and 19. The human blood group O alleles mostly arose from mutations in the A alleles, leading to the inevitable expression of AgH [4,9,26].

GTA and GTB are type II transmembrane proteins containing C-terminal catalytic domains that exist in the lumen of medial Golgi cisternae They have an N-terminal cytoplasmic tail, followed by the trans-membrane domain, the stem region, and the C-terminal catalytic domain located in the lumen of the GC compartments [46]. They are present in the medial GC cisternae localised closer to the trans-pole of the GC. In the epithelium of group A and B individuals, but not in group O individuals, GTA and GTB are observed within the GC trans-side [4,26,47].

After sialylation, proteins go to the APM [39,44]. The non-sialylated polysaccharides go to the BLPM. After sialylation, proteins go to the APM, as they form aggregates due to the formation of hydrogen bonds. The aggregation of polysaccharides is one of the ways to direct proteins and lipids to the APM. When moving towards the post-Golgi APM, the carriers pass through the apical endosomes, where neuraminidase proteins can be located. In endosomes, AgA and AgB are partially de-sialylated. Apparently, during early ontogeny, in enterocytes there is an apical endocytosis. Therefore, the post-Golgi carriers should pass through endosomes in the same way as endocytosed membranes in newborn unfed rats pass through the Golgi apparatus [48].

## 4. Antibodies against AgA and AgB

AbA and AbB are present in the individuals who do not express these antigens (Landsteiner’s Law). Despite the frequency with which ABO(H) antigens and corresponding anti-ABO(H) antibodies are tested clinically [49], surprisingly little is known regarding the factors that drive their development. The ends of linear PSs are not antigenic. Most oligo- and polysaccharides are linear or branched but contain long PS branches. Their immunogenicity is low [4]. Among millions of combinations, only several of them are able to induce the generation of Abs. It seems that short forks consisting of two monosaccharides attached to the same monosaccharide is required. If other monosaccharides are attached to such a fork, then the fork loses its antigenicity. There are, for example, Lewis antigens. Abs directed against carbohydrate antigens exhibit some degree of cross-reactivity. AgA and AgB are abundant and localised at the end of the long oligosaccharide chains which are attached to the heavily glycosylated proteins such as Band-3 or sphingolipids [50,51,52]. It seems that “short forks” on the very ends of PSs are more reactive from the immunology point of view.

The observed affinities of anti-carbohydrate antibodies are typically lower by factors of 103–105 than antibodies specific for protein or peptide antigens [53,54,55]. This is compensated by their initial expression as IgM and their observed class switching bias toward IgG3 in mice and IgG2 in humans, which tend to self-associate through their constant regions to form multivalent networks [56,57,58]. The blood group antibodies are made by CD51 and by CD5-B cells [59].

Newborns may express only 50% of the adult levels of AgA and AgB [51,52]. In newborns up to 4 months old, the titre of AbA and AbB is low [51,52,53,54,55,56,57,58,59,60]. The number of AbA and AbB increases during the first year in humans, as in newborn and young rats and mice. Children below 2 years of age and the elderly respond poorly to polysaccharide antigens. Incompatibility reactions during transfusion begin to manifest themselves only in children 5–10 years old. After 18 years, more and more branched polysaccharides are formed. The titre of AbA and AbB becomes the maximum at the age of 10–20 years, and then, especially in old age, their titre decreases. AbA and AbB mostly belong to the IgM antibodies. Individuals can also make anti-ABO antibodies of the immunoglobulin G (IgG) and IgA isotypes [4,9]. Approximately 40% of the AbA and AbB in individuals with group A and B blood belong to the IgG class. The IgG and IgM antibodies reactive with autologous A and B antigens are present in the normal serum immunoglobulin. The ABO Abs appear to have no natural human target [4,9,59,61]. In individuals of the Bombay type, the AbB titre is higher [9]. In addition, the titre of the antibodies of this group is subject to seasonal fluctuations [50]. Maternal AbA and AbB can cross the placenta and cause haemolytic disease of the foetus and newborn. Individuals with the Bombay phenotype make naturally occurring anti-H antibodies, which are often IgM and IgG [4,26].

## 5. Hypothesis Explaining the Absence of the Target Paradox

Initially, we would like to quote Jajosky et al. [4]. They write: “Conflicting data exist regarding the potential role of plant materials, microbes, genetic elements, and even transient blood group A and B expression, in the development of naturally occurring antibody formation, in general, and formation of anti-ABO(H) antibodies, in particular”. Several hypotheses were proposed for the explanation of this paradox: 1. AbA and AbB are so–called natural Abs. 2. Bacteria could move from the lumen of the intestine into the blood and since they have polysaccharides similar to AgA and AgB, AbA and AbB arise. 3. Viruses carry polysaccharides on their membrane and then they are presented through the auto-presentation system. 4. Glycosylation errors occur due to the overload of enterocytes with lipids. 5. The de-sialylation of PS occurs due to the delay of post-Golgi carriers at the endosome level due to the enterocyte overload with lipids.

## 6. “Natural” Antibodies

The concept posing that AgA and AgB are natural antibodies (NAbs) appeared a long time ago [62,63,64,65,66,67]. Unlike alloantibody formation that occurs following RBC-induced alloantigen exposure [68], NAbs are formed spontaneously (in the absence of foreign Ags) within the first few months of life [69]. Until now, many consensus papers state that anti-ABO Abs form “spontaneously”. However, what this means remains unclear. NAbs are not strictly speaking antigen-specific, yet they contribute to the protection against bacterial and viral infections by polyreactively binding to a wide range of microbes [70]. NAbs were shown to readily form in the absence of antigen exposure, be polyreactive, and be likely of a low affinity [71]. This suggests that anti-ABO(H) antibody formation is not influenced by microbial stimulation. Meanwhile, in normal mice, some bacteria from the gut can cross the gut barrier and appear in the blood every 6 h [72]. NAbs do not undergo somatic hypermutation; a fraction of them may carry mutated variable regions, given that a low rate of hypermutation takes place even without the T-cell signals [73]. Natural antibodies are produced by innate-like B cells, mainly B1 CD5+ cells, upon activation of Toll-like receptors [13].

Isohaemagglutinins (AbA and AbB) were detected in the serum of 79 of 137 (57 per cent) randomly selected full-term newborn infants. AbA and AbB are not of a maternal origin [9]. The infants had isohaemagglutinins of the IgM (γM) or IgG (γG) type. Only two infants had serum γA-globulins. A considerable proportion of the neonatal sera, isohaemagglutinins are of the IgM (γM) type. However, IgG are also present. Therefore, it is concluded that most foetuses produce their own isohaemagglutinins [9]. The maternal IgG antibodies (unlike IgM and IgA) cross the placenta, and maternal and cord IgG autoantibodies showed essentially identical reactivities [74].

The concept of NAbs has several weaknesses. In a letter to the Editor, van Oss [75] criticises Adelman et al. [76] for their concept of so-called natural antibodies. This concept was criticised already in 1959. Indeed, Springer et al. [77] write: “Numerous investigators consider the agglutinins against human erythrocytes of blood group A and B as the prototype of spontaneously originating natural antibodies. On the other hand, antibodies, by definition, are globulins produced or modified in response to the introduction of antigenic substances. This definition clearly is not compatible with the assumption that some of them may be inherited.” Further, other mammals also have their own blood groups (see below). However, there are no such concentrated antibodies against blood groups (their titre is very low) in animals. The reason is not clear. Also, it is unknown why their titre increases after birth. An interesting question is how polysaccharides are processed in endosomes. PSs with AgA and AgB in their ends are quite long. It is difficult for dendritic cells to present on their surface such long structures, especially when they should be attached to the MHC molecule. In order to obtain antibodies of the IgG type with the participation of T lymphocytes and the presentation of peptides and these polysaccharides, sugars are usually cut off from the ends one after another, but since the immunoreactive forks of AgA and Ag are located at the end of PS and even the oligosaccharide, such a mechanism is quite problematic. Figure 2 shows the limited number of highly antigenic PS Ags.

Importantly, animals have their own blood groups. The ABO-like blood groups were found in various mammals, and it is now clear that probably all the mammalian species have the homologue of the human ABO gene [38]. It has been well documented that the foetus produces small amounts of IgM and Ig (γM)-globulins [78]. Mice express an AB-cis transferase capable of generating both the A and B antigen [52]. A human ABO homologue was found within the mouse genome [38]. To date, the three blood groups A, B, and AB have been identified in cats [8,79]. The blood groups of cats are inherited as antigens located on the surface of the membranes of red blood cells, and are determined by special carbohydrates on the membrane of red blood cells, as in humans [80].

The human ABO-like blood group also exists in nonhuman primates [38]. The number of species having both the A and B gene sequences is already rather big; however, it is expected to increase as new genome sequencing projects proceed [21]. Either there are no such natural anti-non-existent antigens in other animals or the titre of the antibodies against non-existent antigens is very low. Therefore, the blood type of an animal is determined only by the antigens of red blood cells present in the body and antibodies agglutinins) are not taken into consideration, namely, the animal blood groups are classified without consideration of the presence of Abs against blood group Ags [81]. What is the difference between animals and humans? Animals never overeat fats and lipids. Experienced cat and dog owners know that pets cannot be overfed with pork and bacon [82].

## 7. Bacterial Hypothesis

According to the bacteria-based hypothesis, based on experiments by Springer et al. [77,83,84] investigating the PS antigens on the bacterial wall, the surfaces of some bacteria contain PS antigens very similar to AgA and AgB. The structural similarity between the bacterial carbohydrate Ags and the surface glycoconjugates of protists, fungi, and animals, leads to the production of carbohydrate-specific antibodies that are protective against a broad range of pathogens [37].

In the situation in which this hypothesis is considered as a paradigm, it is necessary to examine primary data more carefully. The idea of these experiments is based on the well-known fact that chicken have Ab that are able to bind AgA and AgB. The development of carbohydrate-specific antibodies coincides with the microbial colonisation of the gut at birth [4,37]. Thus, if these Abs are a result of the bacteria penetrating into the chicken blood from the gut, the prediction would be the following. The authors decided to keep chicken in germ-free environments. Under these conditions, in the lumen of chicken intestine, bacteria would be absent and the formation of antibodies against AgA and AgB would be blocked. They showed that chickens kept in a germ-free environment would produce anti-B but not anti-A, when fed bacteria expressing high levels of a B-like antigen and lower levels of an A-like antigen. None of the fourteen-day-old chicks showed any anti-human blood group B agglutinins (AbB). The chicks investigated in their study were of the first germ-free generation. Springer et al. [77,83] found anti-B agglutinins to be of higher titre than AbA and AbH. AbAs were found only in 25 to 50 percent of chicks more than 30 days old. The reason for this discrepancy remained unclear.

Later, Springer et al. [83] showed that the bacteria reacted with AbA and AbB. AbA and AbB could be absorbed completely and specifically by E. coli 086, as could anti-A agglutinins by an A active E. freundii, and anti-H(O) agglutinins by S. poona [67,84,85,86]. The distribution of the blood group A, B, and H(O) activity among 282 aerobic Gram-negative bacteria, many isolated from the blood of patients, has been studied. Almost half of these bacteria were found to be blood group-active. About 10 per cent of the organisms exhibited high, disproportional activities, which in some instances approached those of the crude human blood group mucoids. No significant specific D (Rho), M, or N activity was found in approximately 70 members of the Entero bacteriaceae. Blood after incubation with certain bacteria lost the ability to cause precipitation of another group, where there were corresponding antigens. Bacteria inactive in the AbA and AbB inhibition test failed to stimulate the anti-human blood group A, B, or H(O) agglutinins [83].

This hypothesis suffers from several problems. Indeed, Springer et al. [77,83,84] were able to show only a weak production of AbA despite their many studies. The lower titres or absence of AbA can be explained by earlier observations that AgA-like structures could be present in chicks [62,83]. In this case, antibody would be formed after injection of erythrocytes with AgB into the blood of chicks [83]. This had not been tested.

AbA and AbB are detected even in human newborns whose consumption of bacterial polysaccharides is almost impossible. And these antibodies are not of maternal origin [4]. Also, a specific extremely short double-edged “fork” (branches where each branch of two consists of only one monosaccharide) at the tips of bacterial PSs has not been found [23,86]. Springer et al. [77,83,84] used birds and huge doses of bacteria to remove antibodies from the serum [83,84]. Birds have a special immune system; for instance, birds have beta-lysine. Turkeys turned out to be immunologically more reactive than horses, cattle, sheep, and chickens. They have the following main blood groups: A, B, C, D, E, F, G, I, J, K, and L. In birds, the genetic connections of the formation of various antigens are also being studied [81]. In chickens, IgM can be formed with the participation of T lymphocytes. In mammals, this is less pronounced. Also, the bacterial lipopolysaccharides are quite toxic [85]. Only in newborn mammals can enterocytes transcytose pure polysaccharide macromolecules from the intestinal lumen [48,86]. Then, this ability is lost. Bacteria break into the blood, for example, through goblet cells and then cause the synthesis of AbA and Ab. In normal mice, bacteria can be transported into the blood every 6 h [72,87].

Finally, the role of bacteria in the formation of NAbs in mice was not confirmed. On one hand, experiments on germ-free mice stimulated with exogenous antigens are not required for naturally occurring antibody formation. Mice have the AB-cis transferase capable of generating both the AgA and AgB [52]. However, mice have a very small level of spontaneous AbA and AbB formation. The serum IgM repertoire of normal mice is strictly regulated and selected by endogenous ligands [88,89,90]. Large quantities of IgA are constitutively secreted by the intestinal plasma cells. IgAs were not specific to individual bacterial taxa [70]. However, it was shown that the formation of glycan-specific Abs in GalT-KO mice is determined by gut microbiota [91]. This also suggests that anti-ABO(H) antibodies formation is not influenced by microbial stimulation [4,88,89,90,91,92,93].

## 8. Viral Hypothesis

Also, it is possible that AgA and AgB are delivered to the blood of the embryo with viruses. The virus hypothesis suggests that AbA and AbB can be formed as a result of an immune response to the influenza virus (or other viruses), whose epitopes are quite similar to the blood group antigens [4,94,95]. The problem of this hypothesis is the observations demonstrating AbA and AbB existing already in newborns. In embryos, these viruses are not found. If the virus penetrates the endosome, then polysaccharides cannot enter the bloodstream [95]. Importantly, only IgGs can penetrate from the mother’s milk through enterocytes, whereas IgA and IgM cannot [86]. On the other hand, it is well-known that selective pathogenic viruses such as Rubella, Cytomegalovirus, and Herpes simplex virus are able to escape from placental monitoring and thus are capable of vertical transmission from the maternal to foetal tissue [96,97,98,99,100,101,102]. In particular, the enveloped viruses may be effective in perturbing the glycosylation machinery possibly also by mechanisms similar to the lipid overload scenarios. However, the absence of AbA and AbB and other Abs against the blood group polysaccharides in animals also suggests again the virus hypothesis. However, this hypothesis deserves additional analysis.

## 9. Golgi Glycosylation Mistakes

The Golgi mistake hypothesis [94] poses that AgA and AgB are formed by an individual lacking GTA or GTB correspondingly as a result of errors in the synthesis of PSs under conditions when the GC is overloaded with lipids. Under these conditions, there could be errors in the glycosylation of ApoB chylomicrons by enterocytes. For instance, after the overloading of the transcytosis system of Caco-2 cells with lipids, these cells not containing GTA and GTB start to produce AgA and AgB [94]. Caco-2 cells were obtained from a person with group O blood. Due to the absence of both GTA and GTB, these cells should not synthesise AgA and AgB. They synthesise ApoB and ApoE and insert them into chylomicrons. In turn, ApoB, ApoE, and other apolipoproteins have both O and N glycosylation sites [103]. We overloaded the highly differentiated Caco2 cells with an increased concentration of lipids. This led to the appearance of lipid droplets in the cytoplasm, which indicated that the enterocytes were overloaded with lipids. Under these conditions, some of these cells produced AgA. The lipid overloading of enterocytes and cells in other organs induced similar alterations and Golgi tubulation [104,105,106,107].

One of the possibilities to explain these results is to assume that another Golgi enzyme(s) is involved in AgA and AgB synthesis. The glycosyltransferases involved in the biosynthesis of the blood group Ags have redundancy and degeneration. Redundancy is observed when two separate enzymes synthesise the same Ag. For example, fucosyltransferases FUT1, FUT2, and FUT3 are able to synthesise the type 2 carbohydrate H from the same precursor oligosaccharide. Degeneration is when the same enzyme synthesises different polysaccharide structures. For example, fucosyltransferase FUT2 is able to synthesise carbohydrates of type 1 and type 2 h. From their respective oligosaccharide precursors of type 1 and type 2, and fucosyltransferase 3 is capable of collecting at least four different carbohydrates of the blood histo-groups. Fucosyltransferase-3 is capable of synthesising at least four different groups of carbohydrates in the blood [4,6].

Also, it is well-known that in the GC, the synthesis of PSs occurs with low preciseness [108]. There is a significant overlap of the functional characteristics between different GGEs [108]. Some GGEs could make the job of other enzymes, namely, by performing the work that was previously considered specific to other enzymes. Instead of monosaccharides being specific to this enzyme, they can attach not their specific sugar, but the opposite one [26,109]. For instance, in pigs lacking galactosyltransferase (GGTA1), another transferase, the GGTA2, remains and can produce α-Gal antigens. Also, in pigs, after elimination of one of the Golgi enzymes, the necessary polysaccharide was synthesised by another enzyme [110,111]. Importantly, a weak AbB titre is present even in people with AgB [9]. Interestingly, people with group A or B blood can produce antibodies against AgA, but the opposite situation never happens [25]. There is considerable variability in the oligosaccharide and PS chains even in the presence of a particular gene [25]. In cell culture, after GC cargo overload, the synthesis of the sugar chains occurs with a low accuracy, inducing a change in glycosylation [112,113]. The “short forks” could have two consecutive fucoses [4,25] (see also Table 2 presented in [114]).

There is considerable variability in PS chains even in the presence of a particular gene [25]. “Short forks” could have two consecutive fucoses [4,25] (see also Table 2 in [115]). In addition, the GTA2 has a lower activity and has a narrower substrate specificity compared to the GTA1. Mutant alleles of the corresponding FUT1 and FUT2 genes lead either to the H–phenotype (Bombay phenotype) or to a weak H-phenotype [12]. Because of a very small difference between GTA and GTB, some variant enzymes can simultaneously create both AgA and AgB [115]. For instance, missense mutations outside the catalytic domain of the ABO glycosyltransferase can cause weak blood group A and B phenotypes. Moreover, because GTA and GTB are not 100% efficient, blood group A, B, and AB individuals also express some AgH [116,117].

Glycosylation depends not only on the presence of glycosyltransferases in cells, but also on their localisation in the GC; the availability of nucleotide sugar transporters, which should localise nearby, as donors of nucleotide sugars are necessary for their synthesis; the final destination of the glycoprotein or lipid; as well as the action of decomposing glycosydases that cleave glycans. The structure of the GC varies significantly depending on the functional state and cell phase [113,118,119,120]. Although GTA and GTB are mostly located at the trans-side of the medial GC in goblet cells, these enzymes were found throughout the Golgi stack in enterocytes containing microvilli [47]. The altered trafficking of variant ABO transferases is also involved in the formation of weak ABO phenotypes [116,119].

Thus, it is not possible to think that some variant GGEs can simultaneously create both AgA and AgB in ApoB. When enterocytes are overloaded with lipids, they begin to make mistakes and the enzymes that usually cannot synthesise AgA and AgB begin to synthesise them in ApoB. All these observation suggest that the GGEs can easily make glycosylation “mistakes”. So, Abs against AbA or AbB can be synthesised under certain conditions, namely, after overloading the GC of enterocytes during a meal containing a lot of lipids, AgA and AgB may occasionally appear in the blood, causing the production of the corresponding antibodies. Already the first feeding represents a significant problem for the enterocytes [85].

An interesting question is why in some human embryos there are already AbA or AbB whereas in these organisms GTA or GTB are absent. Why do newborns have antibodies? The explanation could be the following. Due to the rapid growth of cells and intensive synthesis of the membranes necessary for cell division, glycosylation errors are possible. Also, the foetus swallows amniotic fluid, but to varying degrees. In newborns, after endocytosis of the membranes from amniotic fluid, they pass through the GC [48]. As a result, glycosylation errors in the enterocyte GC could also occur during embryonic development. This hypothesis explains why animals have very low titres of AbA and AbB (of course, against their own Ags). According to this concept regarding the GC errors, human subjects eat too many lipids whereas animals do not. This explains why animals have no antibodies against their blood group Ags, which do not exist, similarly to the situation with AbA and AbB in humans.

## 10. De-Sialylation of AgA and AgB in Endosomes

According to the concept of AgA and AgB de-sialylation, AgA and AgB can be synthesised with a usual Ag fork at the end and with a fork covered with sialic acid. It is known that sialic acid inducing the aggregation of PSs via hydrogen bonds provides the transport of proteins from AgA and AgB to the APM [39,44,119,120]. This is especially important in endothelial cells, where a high concentration of AgA and AgB is present mostly on the APM [120]. When enterocytes are overloaded with lipids, chylomicrons (containing AgA and/or AgB) stagnate at the endosome level, where neuraminidases are localised [94]. Post-Golgi carriers often form membrane contacts and even connections with endosomes [39,44,48,120]. We have shown that when the enterocytes of the small intestine are overloaded with lipids, chylomicrons, which consist of lipids and ApoB, can stagnate during transport in the post-Golgi compartment [48,107], which could temporarily connect with endosomes where de-sialylation could occur. There are four types of mammalian sialidases: Neu1–Neu4 [121].

It is possible to assume that under normal conditions, some AgA and AgB could be masked by sialic acid and do not cause an immunological reaction. Interestingly, nobody has exhaustively demonstrated that AbA and AbB are synthesised without the addition of sialic acid to these fork monosaccharides. When passing through the endosomes, AgA and AgB can be processed by neuraminidases1–4, which are localized there. The enlarged chylomicrons were stopped at the level of the LAMP2 and Neu1 positive post-Golgi structures, secreted, fused, delivered to the interstitial space, captured by the lymphatic capillaries, and transported to the lymph node, inducing the movement of macrophages from the lymphatic follicles into its sinuses [48]. If sialylation occurs, then proteins and lipids go to the APM, and, passing through the endosomes, undergo de-sialylation. After removal of sialic acid, AgA and AgB could be exposed. For instance, it is known that the final form of transformation of chylomicrons in the blood is LDL. Previously, it was shown that the so-called neutral LDL found in the blood of patients with atherosclerosis [122] contained a reduced amount of sialic acid residues. These particles exhibited proatherogenic properties [122,123,124].

On the other hand, it is shown that in erythrocytes of group O, a lot of Ag is present without its conversion into AgA and AgB. Also, it is possible to find GalNAc attached to the end of AgB [25]. The cis-AB phenotype can raise questions about an apparently paradoxical inheritance of the ABO blood group, such as the birth of an O child from an AB mother. AgH type 0 could be transformed into Lewes-b/y; AgA into A-Lewis, AgB into B-Lewis as a result of the fucose attachment to GlcNAc [4]. Cis-AB, a rare ABO variant, is caused by a gene mutation that results in a single glycosyltransferase enzyme with dual A and B glycosyltransferase activities [125]. Finally, glucose could be attached to fucose [58]. The post–Golgi carriers could pass through endosome [126]. If antigenic “short forks” masked with sialic acid are retained longer in the endosomal compartment they can be treated with Neu-4. As a result, the sialic acid masking the antigenic fork would be removed by this neuraminidase. Alternatively, one could assume that glycosylation mistakes are constantly being made, but as long as there is no de-sialylation of AgA and AgB, the erroneously synthesised ones are not visible to antibodies.

## 11. Blood Groups and Immune Tolerance: How Do Our Hypotheses Correlate with the Results of Studies on Immunological Tolerance?

The glycosylation error and de-sialylation hypotheses are close to the concept of natural antibodies. In order to make our hypothesis clearer, we will briefly describe here the current view on tolerance to auto-antigens. Antigens are divided into T-lymphocyte-dependent (TD) and T-lymphocyte-independent (TI) Ags. Proteins and peptides are usually TD-antigens, since in order to trigger an immune response, their stimulation by helper T-lymphocytes is necessary. The molecules of the main histocompatibility complex (MHC), which are transported to the plasmalemma in macrophages, dendritic, and some other cells. Macrophages and dendritic cells present TDAg to the T-lymphocyte. At the same time, due to the formation of memory B- and T-lymphocytes, a prolonged immune response is induced. The Abs against TDAgs have a high affinity for Ags and have several isotypes (IgA, IgM, IgG1, IgG2a, IgG2b, IgG3). Most PS antigens are TIAgs. Unlike TDAgs, TIAgs do not lead to the formation of immunological memory [127,128]. TIAgs can induce proliferation and differentiation of both naive and mature B lymphocytes [23,127,128].

In order to connect our error hypothesis with the concept of natural antibodies, it is necessary to assume that due to the low number of possible combinations of sugars, the immune system contains clones of all such combinations. Indeed, carbohydrate-specific NAbs arise without prior exposure to exogenous antigens. In contrast to the antigen-specific antibodies, which are produced in a T-cell-dependent manner by mature B cells, natural antibodies are defined as pre-immune antibodies, generated without antigenic stimulation and T-cell assistance [13]. Usually, these chains of PSs have a rather low immune reactivity. Only a few domains of PS chains could induce a strong immune reaction. Just before birth, Abs against these domains start to be synthesised. The tolerance of a host to its own PS molecules is acquired during the late stages of embryogenesis and immediately after birth [129,130,131,132]. Abs could be delivered to newborns in the milk of their mothers [133]. During the first year of life, or already in the embryo, the clones of the lymphocytes reactive to AgA and AgB, which are synthesised in the organism, are eliminated in the thymus through cell apoptosis. Only lymphocyte clones, which cannot bind PSs generated by the host remain. In persons without GTA or GTB, the lymphocyte clones against AgA or AgB are preserved. Due to glycosylation mistakes or de-sialylation, these Ags can appear. The clones of lymphocytes specific for these Ags were not eliminated after the birth and are ready to react forming AbA or AbB or both. Due to the peculiar nature of human food (a lot of lipids), these Ags are formed from time to time [129,130,131,132,134,135,136]. This induces the augmentation of their titres. Importantly, this mechanism (glycosylation errors or de-sialylation) could also explain autoimmune diseases.

Also, we cannot exclude that the synthesis of carbohydrate-specific antibodies may in part follow the classical antigen presentation pathway and T-cell-dependent activation. Glycopeptides can be presented via MHC-II like standard peptide antigens. The carbohydrate moiety can be recognised by glycan-specific B cells, while T cells specifically recognising the same glycopeptide antigens provide the necessary co-stimulatory activity ensuring antibody maturation [137,138]. Some zwitterionic oligosaccharides and PSs devoid of peptide components could be processed by MHC-II to activate T cells and B cells [139,140,141,142].

## 12. Conclusions and Future Perspectives

The combination of the natural antibody concept with the hypothesis that augmentation of the concentration of AbA and AbB could depend on food content seems plausible. All these hypotheses should be tested based on the predictions they formulate. There are some predictions derived from these new hypotheses. Indeed, the titre of agglutinins will be higher in those who eat a lot of fatty foods, since antigens A and B, formed due to Golgi errors in enterocytes, react with the existing AbA and AbB and reduce their titre. It is also known that AbA and AbB increase the risk of rejection mediated by these antibodies during solid organ transplantation [6]. The organs with weakly polarised ECs of blood capillaries (such as the kidneys and liver, but not the heart and lung) should be more sensitive to transplantation rejection, even between people with incorrectly matched blood groups, since due to their lower polarity, AgA and AgB can diffuse to the lumen plasma membrane of weakly polarised ECs. This could be a risk factor for the development of delayed graft function [137]. An additional test is to feed mice with a large amount of lipids and see if AbA and AbB accumulates or not. In any case, it is necessary to compare them. The problem is that after a hypothesis becomes consensual, it could be very difficult to convince its followers, even if the alternative hypothesis looks more solid and resolves all the contradictions.

## Figures and Tables

**Figure 1 ijms-24-15044-f001:**
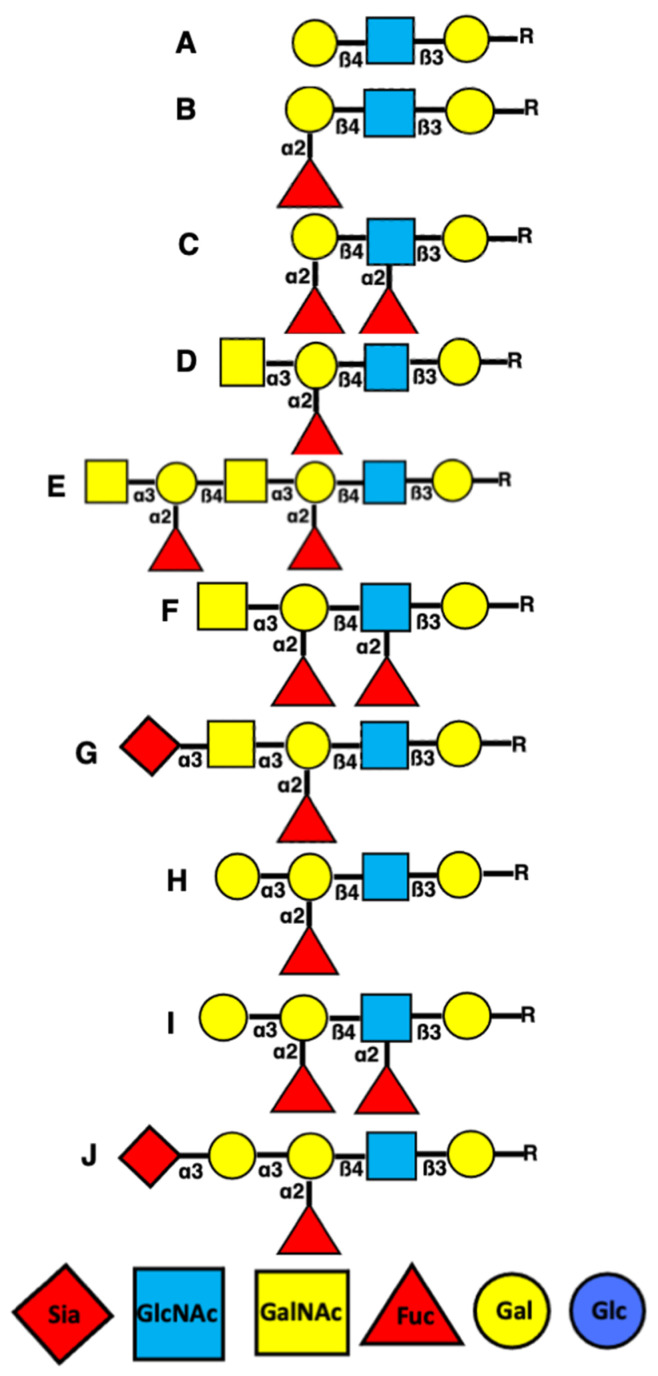
Scheme of different forms of blood group antigens (A, B, D, E, G). (C, F, I) Lewis Ags formed after attachment of fucose to GlnNAc from H (shown in plate B), A (shown in plate D), and B (shown in plate H) type Ags. (A) H-Null Ag. (B) H-Ag; type O. (C) Lewis b/y Ag. (D) A-Ag; type A1. (E) A-Ag; type A2. (F) A-Lewis Ag. (G, J) The scheme of antigen masking by sialic acid. Possible appearance of the ends of AgH, AgA, and AgB after sialylation. (H) B-Ag; type B. (I) B-Lewis Ag. Abbreviation: α3 indicates α1–3; β3 corresponds to β1–3; β4 corresponds to β1–4.

**Figure 2 ijms-24-15044-f002:**
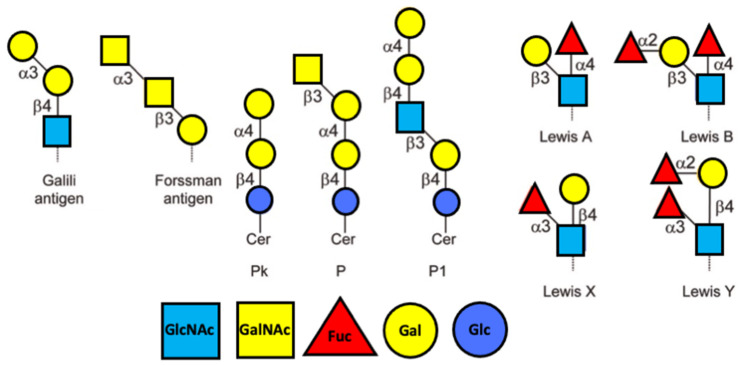
Schemes of highly antigenic domains of PS, namely, Galii Ag; Forssman Ag; Lewis X and Y Ags, P blood group Ag Pk; P, and P1. α3 indicates α1–3; ß3 corresponds to ß1–3; ß4 corresponds to ß1–4. Sugars are indicated in the scheme below.

## Data Availability

Not applicable because the paper is the opinion based on the analysis of the piublished literature.

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
