# Peer review of "Mechanisms of Formation of Antibodies against Blood Group Antigens That Do Not Exist in the Body"

_ijms, 2023, doi:10.3390/ijms242015044_

Round 1

Reviewer 1 Report

This is a nice scientific opinion article that puts the spotlight on the long standing paradoxical observation that humans can form antibodies against blood group antigens that do not exist in their body. The article investigates and critically discusses several hypotheses that could offer a plausible explanation for this interesting phenomenon. Most of the different hypotheses are discussed in detail and enable the reader to understand the scientific context , background and relevance of the topic.

A minor shortcoming of the article is that the virus hypothesis has been getting to little attention. It deserves a far more diligent discussion as it seems to offer at least a similarly plausible explanation as some of the other hypothetical scenarios presented in the article. It is well known that selective pathogenic viruses such as Rubella, Cytomegalvirus and Herpes simplex virus are able to escape from placental monitoring and thus are capable of vertical transmission from maternal to fetal tissue. Especially, enveloped viruses may be effective in perturbing the glycosylation machinery possibly also by mechanisms similar to lipid overload scenarios.

Figure 3 is not referenced in the text. It should be mentioned there as well so that the figure does not stand alone.

The English language in the text is of varying quality. While some passages seem to be written by a native or business fluent speaker other sections are riddled with mistakes. Therefore, the article requires major proofreading and editing prior to publication.

Author Response

Reviewer 1

This is a nice scientific opinion article that puts the spotlight on the long standing paradoxical observation that humans can form antibodies against blood group antigens that do not exist in their body. The article investigates and critically discusses several hypotheses that could offer a plausible explanation for this interesting phenomenon. Most of the different hypotheses are discussed in detail and enable the reader to understand the scientific context , background and relevance of the topic.

Reply: We acknowledge our reviewer for kind word. The text in PDF format is presented without indication of the text changers.

A minor shortcoming of the article is that the virus hypothesis has been getting to little attention. It deserves a far more diligent discussion as it seems to offer at least a similarly plausible explanation as some of the other hypothetical scenarios presented in the article. It is well known that selective pathogenic viruses such as Rubella, Cytomegalovirus and Herpes simplex virus are able to escape from placental monitoring and thus are capable of vertical transmission from maternal to fetal tissue. Especially, enveloped viruses may be effective in perturbing the glycosylation machinery possibly also by mechanisms similar to lipid overload scenarios.

Reply: We added a new portion of the text on this issue.

Figure 3 is not referenced in the text. It should be mentioned there as well so that the figure does not stand alone.

Reply: We eliminated Figure 3.

The English language in the text is of varying quality. While some passages seem to be written by a native or business fluent speaker other sections are riddled with mistakes. Therefore, the article requires major proofreading and editing prior to publication.

Reply: We corrected our English

Reviewer 2 Report

Find the comments in the attached file.

The language has to be improved significantly. There are MANY mistakes. For examples see attached file.

Author Response

Reviewer 2.

Comments on the Quality of English Language. The language has to be improved significantly. There are MANY mistakes. For examples see attached file.

Reply: We corrected our English. The text in PDF format is presented without indication of the text changers.

Even if the authors did not convince me with their argumentation, their arguments and points of view should be presented to the scientific community for further discussion.

Reply: This phrase suggests that this reviewer is a real scientists. I  got such reasoning for the first time in my life. Thanks a lot.

Currently the text is very hard to read.

Reply: We improved the text and made it simpler.

There are too many unexplained abbreviations and the quality of language is weak.

We eliminated a lot of abbreviations and added explanation.

Reply: The English was corrected.

There are several redundant paragraphs

Reply: We eliminated some part of the text.

On the other hand, some statements are not explained.

Reply: We explained these parts.

Many typos

Reply: We corrected them.

Inhomogeneous nomenclature

Reply: We made unification of terms.

p.4: in rare cases NauGc may also occur in human tissues! doi: S0021-9258(19)81387-X, doi: 10.2183/pjab.82.181, doi: 10.1073/pnas.2131556100

Reply: We corrected this mistake and added corresponding references.

p.5: Type II membrane proteins contain a cytoplasmic tail, then the trans-membrane domain followed by the stem region, ... not the other way round. The stem region is inside of the vesicle.

Reply: We corrected this mistake.

p.6: "Since the number of combinations of monosaccharides is much smaller than the number of combinations of amino acids... " This statement is wrong.

Reply: We corrected this mistake and eliminated this statement.

Rephrase the conclusion and focus more on the home message you want to give.

Reply: We rewrote the conclusion part.

Too many (not necessary) abbreviations, which sometimes only explained somewhere in the text or sometimes not even there: GGE, APM, EC, GTF, VH, and others.

Reply: We eliminated some abbreviations and added all necessary explanations for them.

Sometimes ABO, sometimes AB0

Reply: We changed AB0.

Many typos such as: p.3 ABA instead of AbA, desialilation instead of desialylation, imm50/unology, dearly instead of clearly, several time AbA or AbB in incomplete – just Ab, gly-cosyltransferase.

Reply: We corrected these mistakes.

Missing something: p.2: "naturally occurring...."

Reply: We eliminated this sentence.

P.3: Antigens of blood groups of ABO system are carbohydrates

Reply: We corrected this sentence.

  1. 5: The exception is collagens.

Reply: We eliminated this phrase.

p.7: Animals blood group

Reply: We corrected this sentence.

The interesting question...

Reply: We corrected this mistake.

p.5: Usually Fucose attaches to galactose ...

Reply: We corrected this mistake.

  1. 6. Bacteria...

Reply: We corrected this mistake.

p.7: It is also unknown...

We corrected this mistake.

Unclear wording of the following sentences...

Reply: We corrected or eliminated these sentences.

Figure 1. Change it according to Fig. 2...

Reply: We changed it.

Figure 3. ...

Reply: We eliminated this Figure according to the demand of the first reviewer.

Editor suggestions

We encourage all the scholar to consider this and reduce the
self-citations to make sure that only the most relevant citations are
kept. The self-citation rate should not exceed 15%.

Reply: Now this level is below 15%. Also, we eliminated mots of text similar to those in the papers indicated by the editor,

We encourage you to review the level of English and decide
whether revisions are necessary. If the reviewers do not provide detailed
comments, you may ask the assistant editor to contact the reviewers for more
specific comments.

Reply: We corrected our English.

Round 2

Reviewer 2 Report

Comments are attached

There are still many many mistakes.

Author Response

Reply

We thanks our reviewer again and again for his (her) real scientific position.

  1. We corrected the chapter 8 Bacterial hypothesis. The new text is red.

  1. We modified Fig. 1 according to Fig. 2. Also, we indicate α2 orientation similarly as in Galili antigen; Ok; P, P1; Lewis B in Fig 2.

  1. We have added references indicating that NeuGc is also used already in the previous version of the paper.

  1. We corrected the sentence suggesting that AGA AgB are PS. Now the text (in red) indicates that they are oligosaccharides.

  1. We removed additional spaces.

  1. We corrected and now did not find unnecessary [[, ]], ((, and )).

  1. We replaced our with own.

  1. (b) was eliminated.

  1. Are was eliminated.

  1. Depends was replaced with depend.

  1. We replaced have into has.

  1. We eliminated unnecessary nomenclature information.

  1. We eliminated to be in p13.

  1. We corrected all problems related to readability and language.